# Inhibitors of the Oncogenic PA2G4-MYCN Protein-Protein Interface

**DOI:** 10.3390/cancers15061822

**Published:** 2023-03-17

**Authors:** Hassina Massudi, Jie-Si Luo, Jessica K. Holien, Satyanarayana Gadde, Sukriti Krishan, Mika Herath, Jessica Koach, Brendan W. Stevenson, Michael A. Gorman, Pooja Venkat, Chelsea Mayoh, Xue-Qun Luo, Michael W. Parker, Belamy B. Cheung, Glenn M. Marshall

**Affiliations:** 1Children’s Cancer Institute Australia for Medical Research, Lowy Cancer Research Centre, UNSW Sydney, Sydney, NSW 2750, Australia; 2Department of Paediatrics, The First Affiliated Hospital, Sun Yat-sen University, Guangzhou 510060, China; 3School of Science, STEM College, RMIT University, Melbourne, VIC 3000, Australia; 4ACRF Rational Drug Discovery Centre, St. Vincent’s Institute of Medical Research, Fitzroy, VIC 3065, Australia; 5ACRF Facility for Innovative Cancer Drug Discovery, Bio21 Molecular Science and Biotechnology Institute, The University of Melbourne, Parkville, VIC 3010, Australia; 6School of Women’s and Children’s Health, UNSW Sydney, Sydney, NSW 2750, Australia; 7Kids Cancer Centre, Sydney Children’s Hospital, Randwick, NSW 2031, Australia

**Keywords:** neuroblastoma, PA2G4, MYCN, inhibitors, WS6

## Abstract

**Simple Summary:**

Neuroblastoma is the commonest solid tumour of early childhood. Most children presenting with advanced disease have a poor survival rate, despite intensive chemotherapy treatment. One third of advanced NB tumours have MYCN oncogene amplification and overexpression, which predicts poor patient prognosis. However, the development of inhibitors directly targeting MYCN family proteins has proven challenging due to the absence of MYCN structural information. We have identified a protein called PA2G4, which is a cofactor for MYCN in promoting cancer cell growth. We have developed a compound WS6 and its analogous, which inhibits PA2G4 and MYCN protein levels and reduces tumour cell growth. The purpose of this study is to characterise and evaluate our novel WS6 analogous as inhibitors of MYCN and PA2G4 oncogenic functions in neuroblastoma.

**Abstract:**

MYCN is a major oncogenic driver for neuroblastoma tumorigenesis, yet there are no direct MYCN inhibitors. We have previously identified PA2G4 as a direct protein-binding partner of MYCN and drive neuroblastoma tumorigenesis. A small molecule known to bind PA2G4, WS6, significantly decreased tumorigenicity in *TH-MYCN* neuroblastoma mice, along with the inhibition of PA2G4 and MYCN interactions. Here, we identified a number of novel WS6 analogues, with 80% structural similarity, and used surface plasmon resonance assays to determine their binding affinity. Analogues #5333 and #5338 showed direct binding towards human recombinant PA2G4. Importantly, #5333 and #5338 demonstrated a 70-fold lower toxicity for normal human myofibroblasts compared to WS6. Structure-activity relationship analysis showed that a 2,3 dimethylphenol was the most suitable substituent at the R^1^ position. Replacing the trifluoromethyl group on the phenyl ring at the R^2^ position, with a bromine or hydrogen atom, increased the difference between efficacy against neuroblastoma cells and normal myofibroblast toxicity. The WS6 analogues inhibited neuroblastoma cell phenotype in vitro, in part through effects on apoptosis, while their anti-cancer effects required both PA2G4 and MYCN expression. Collectively, chemical inhibition of PA2G4-MYCN binding by WS6 analogues represents a first-in-class drug discovery which may have implications for other MYCN-driven cancers.

## 1. Introduction

Neuroblastoma is the most common solid tumour in early childhood [1]. Almost 50% of children with neuroblastoma present with clinically advanced disease and, despite multimodal therapy, survival is poor with treatment options being highly toxic. The v-myc avian myelocytomatosis viral oncogene neuroblastoma derived homolog (MYCN) is a common cancer driver gene in the most aggressive form of neuroblastoma [2] and other cancers, such as medulloblastoma [3], ovarian [4], and prostate cancer [5]. Aberrant high MYCN expression and amplification is a major clinical determinant of patient relapse [1].

The development of direct inhibitors of MYCN proteins, thus far, has been difficult. The MYC protein family, of which MYCN is a member, are intrinsically disordered proteins with no deep pockets for drug design. A variety of MYCN protein functions are dependent on protein-protein interactions [6]. Extensive evidence from us and other researchers has shown that some MYCN binding proteins markedly increase MYCN protein stability to drive tumour formation [7,8,9]. Inhibition of MYCN protein binding partners can markedly reduce MYCN stability and is an emerging treatment strategy for MYCN-driven cancers [10,11].

We have previously shown that proliferation-associated protein 2G4 (PA2G4) is a MYCN binding protein that stabilised MYCN in a positive feed-forward expression loop [11]. WS6, a chemical inhibitor of PA2G4, disrupted the PA2G4-MYCN binding in neuroblastoma cells with significant antitumour effects in vitro and in vivo [11]. By disrupting the PA2G4-MYCN interaction, treatment with WS6 increased MYCN degradation, in turn decreasing overall levels of MYCN and PA2G4 protein in neuroblastoma cells. However, WS6 exhibited a narrow therapeutic window, with significant toxicity for normal fibroblasts in vitro and in perinatal mice. Furthermore, WS6′s physicochemical properties, in particular its high molecular weight (656.74 Da) and clogP (7.31), suggest it may have significant off-target effects, which preclude clinical development [12].

Small molecule inhibition of protein-protein interfaces is a challenge due to the undruggable nature of a protein-protein interface [13]. However, the advantages of small molecules have led to research into the ideal physicochemical properties of successful protein-protein interaction inhibitors [14,15,16]. Our recent analysis showed that, although protein-protein modulators on the market tend to have more variability in their physicochemical properties, those in clinical trials more frequently demonstrate traditional drug-like parameters [14]. This study indicated that protein-protein inhibitor hit compounds should aim for a cLogP < 3.5, molecular weight < 500, total polar surface area < 100, hydrogen bond donors < 3, and rotatable bonds < 6.

In order to increases WS6 drug-like properties, we studied the cytotoxic effects of different WS6 analogues, with improved physicochemical properties. Two of our hit compounds, #5333 (also called SVI-5333) and #5338 (also called SVI-5338), had a significantly increased therapeutic window compared to WS6 and inhibited the malignant phenotype of neuroblastoma cells in vitro, which partly depended on PA2G4 and MYCN expression. Quantitative mass spectrometry revealed numerous differentially regulated protein targets of #5333 and #5338, providing insights into functional pathways and off target activities, as well as potential biomarkers of the drug response.

## 2. Materials and Methods

### 2.1. 2D Analogue Search

A virtual screen of our in-house library of ~16.5 million commercially available compounds was conducted with UNITY (Sybylx2.1), using a backbone of the previously identified compounds and a minimum 80% Tanimoto similarity index [17]. All compounds were purchased from Chemdiv (https://www.chemdiv.com/).

### 2.2. Molecular Docking

The human PA2G4 crystal structure (PDB: 2Q8K) was retrieved from the PDB [18]. All waters and crystallisation ligands (glycerol and sulfate ions) were removed, and hydrogen atoms were added in Sybylx2.1 (Sybylx2.1, Certara L.P, St. Louis, MO, USA). The Geom-Dock (Sybylx2.1) software function was used to create a protomer (docking receptor) with a bloat protomer level of 2 Å (default 0) and threshold reduced to 0.43 (default 0.50). Three-dimensional (3D) conformers of compounds were created in GeomX-Dock (Sybylx2.1), with parameters set to allow flexibility of rings and movement of heavy atoms and hydrogens. The starting conformations were increased to 5 (default 0) and the number of poses to optimise was set at 20 (default 10). All other parameters were left at the default setting. The resultant dock was analysed visually, and clusters were identified for each ligand.

A second program, Fast-Rigid Exhaustive Docking (FRED) (OpenEye Scientific Software, Santa Fe, NM, USA), was used to confirm the binding mode. A docking receptor was created using the PA2G4 structure (PDB: 2Q8K) and the Make Receptor program (OpenEye Scientific Software, Santa Fe, NM, USA) [19]. OMEGA v2.4.3 was then used to create a multi-conformer database of WS6 and the analogues. FRED was used to perform a rigid binding pose exploration for PA2G4 protein and ligand complexes. Poses were ranked using the Chemgauss4 scoring function as an indication of pose and docking quality [19,20]. FRED docking results were then visualised using VIDA (OpenEye Software, Santa Fe, NM, USA) to identify conformational clusters for each ligand.

### 2.3. Cell Lines

The MYCN amplified neuroblastoma cell line SK-N-BE(2)-C was kindly provided by Professor June Biedler (Memorial Sloan Kettering Cancer Centre, New York, NY, USA), and Kelly was obtained from European Collection of Cell Cultures. The two normal lung fibroblast cell lines (MRC-5 and WI-38) were purchased from ATCC. SK-N-BE(2)-C shMYCN#2 doxycycline-inducible cell and Kelly shMYCN#2 doxycycline-inducible cell were custom-made by GenScript (Piscataway, NJ, USA). SK-N-BE(2)-C, SK-N-BE(2)-C shMYCN#2, and SHEP MYCN3 doxycycline-inducible cells were maintained in Dulbecco’s modified Eagle’s medium (DMEM) (Invitrogen, Life Technologies, Waltham, MA, USA) with 10% foetal calf serum (FCS). Kelly and Kelly shMYCN#2 doxycycline-inducible cell were cultured in Roswell Park Memorial Institute (RPMI) 1640 (Invitrogen, Life Technologies) with 10% FCS. MRC-5 and WI-38 were cultured in Alpha-MEM media (Invitrogen, Life Technologies) with 10% FCS. All cells were not cultured for more than 2 months. All used cell lines were authenticated by Cell Bank Australia (Westmead, NSW, Australia) as being free from mycoplasma, and cultured at 37 °C and 5% CO_2_ in a humidifier incubator.

### 2.4. Protein Production

PA2G4 protein construct was ordered from GenScript (Piscataway, NJ, USA) and cloned into a pET22b vector. The construct was expressed in BL21 (DE3) *E. coli*, which was transformed by adding 2 μL of DNA, as per the Addgene Bacterial Transformation Protocol (Addgene, Cambridge, MA, USA). A total of 250 μL of transformed cells were grown in Luria-Bertani (LB) agar, containing 100 μg/mL of ampicillin, overnight at 37 °C. Starting cultures were created by inoculating 100 mL of LB with several bacterial culture colonies, followed by an overnight shaking incubation at 125 rpm and 37 °C. The overnight culture was added (10 mL/L) to 2 L flasks of LB and incubated at 125 rpm and 37 °C until log phase growth was reached (OD600 = 0.6). Following incubation, the cell line was centrifuged at 4200 rpm for 30 min in 1 L bottles. The cell pellet was then resuspended in 25 mL of phosphate-buffered saline (PBS) and centrifuged again at 4200 rpm for 30 min.

Cell pellets were resuspended in 25 mL of Ni-IMAC buffer A (100 mM Tris (pH 8.0), 0.5 M NaCl, 1 mM TCEP), filtered through 22–25 μm Miracloth, and placed on ice. Filtered cells were lysed using the EmulsiFlex-C5 High-Pressure homogenizer (Avestin, Ottawa, ON, Canada), followed by high-speed centrifugation at 18,000 rpm for 20 min. The supernatant, containing the soluble cell lysate, was loaded onto a 5 mL His-Trap column (GE Healthcare Life Sciences, Buckinghamshire, UK) equilibrated in Ni-IMAC buffer A. An incremental gradient of Ni-IMAC buffer B (100 mM Tris pH 8.0, 0.5 M NaCl, 1 mM TCEP, and 1 M imidazole) was introduced. Eluted fractions were analysed by SDS-PAGE and pooled accordingly. Each protein sample was then loaded onto a HiPrep 26/10 Desalting column (GE Healthcare Life Sciences, Buckinghamshire, UK) and equilibrated into anion exchange buffer A (100 mM Tris pH 8.0, 50 mM NaCl, 1 mM TCEP). Samples were loaded onto the column and eluted fractions were analysed by SDS-PAGE and pooled accordingly.

Protein was loaded onto a 5 mL HiTrap Q (Sephrose-Q) anion exchange column (GE Healthcare Life Sciences, Buckinghamshire, UK) and equilibrated in anion exchange buffer A. Samples were loaded, eluted in the flow-through, and analysed using SDS-PAGE. Samples were pooled and concentrated using 10 kDa MWCO Ultra-4 centrifugal filters (Merck, Macquarie Park, NSW, Australia) and intermittent centrifugation at 4000 rpm. Protein samples were then purified using Size Exclusion Chromatography (SEC). Each sample was concentrated to 10 mL using MWCO Ultra-4 centrifugal filters, then loaded onto a Superdex 75 16/60 SEC column (GE Healthcare Life Sciences, Buckinghamshire, UK) and equilibrated in SEC buffer (25 mM Hepes pH 7.7, 100 mM NaCl, 1 mM TCEP). Samples were eluted, analysed by SDS-PAGE, and concentrated to 10 mg/mL using 10 kDa MWCO Ultra-4 centrifugal filters, with intermitted centrifugation at 4000 rpm.

### 2.5. Surface Plasmon Resonance (SPR)

Recombinant PA2G4 was buffer exchanged into PBS buffer using an overnight dialysis at 4 °C. Protein was concentrated to 0.5 mg/mL and incubated for 3 h on ice with a 20:1 molar excess of EZ-Link Sulfo-NHS-LC-Biotin (Thermo Scientific, Melbourne, VIC, Australia). Biotinylated protein samples were isolated from excess biotin using a Superdex 200 10/300 SEC column (GE Healthcare Life Sciences, Buckinghamshire, UK). All SPR experiments were performed on a Biacore T200 instrument (GE Healthcare Life Sciences, Uppsala, Sweden), at 25 °C, in the presence of HEPES Buffered Saline (HBS) (20 mM phosphate (pH 7.5) 137 mM NaCl, 2.7 mM KCl, and 0.05% Tween 20). Streptavidin was coupled to a CM5 chip (GE Healthcare Life Sciences, Buckinghamshire, UK) using amine coupling. Biotinylated protein samples were then capture-coupled to the streptavidin, with flow cell 1 blocked as a reference.

Compounds were dissolved in 100% DMSO and diluted to 10 mM. Prior to conducting compound SPR, a DMSO solvent correction was performed following the Biacore Laboratory guideline 29-0057-18 (GE Healthcare Life Sciences, Buckinghamshire, UK). Compounds were injected at 30 μL/min for 60 s across the chip in a 3-fold dose-response manner. SPR results were analysed using the Biacore T200 Evaluation Software v.1 [21].

### 2.6. Colony Formation Assays

A total of 250 SK-N-BE(2)-C and 500 Kelly cells were seeded in 6-well plates and treated with #5333 and #5338 24 h after seeding. Once colonies had formed (14 days for Kelly, 10 days for SK-N-BE(2)-C), cells were fixed and stained with 0.5% crystal violet in 50% methanol and washed twice to remove the unincorporated stain. The photos of colonies were taken by ChemiDoc MP Imaging System (Bio-Rad, South Granville, NSW, Australia). Stained colonies were counted, and colony size was measured using Image J software.

### 2.7. Cell Proliferation Assays

Cells were seeded in 96-well plates the day before and then treated with #5333 and #5338 for 72 h. Cell proliferation was measured using the BrdU ELISA kit (Roche, Millers Point, Australia), according to the manufacturer’s instructions. 10 μM BrdU was added to media and incubated at 37 °C in 5% CO2 for 2 h. Cells were fixed with supplied fixing solution for 30 min and then blocked with 10% FCS in PBS for 10 min. Peroxidase-conjugated anti-BrdU (1:100) was added and incubated for 90 min at room temperature. After washing with PBS for three times, peroxidase substrate was added to the cells. Changes in cell proliferation were calculated from the absorbance readings at 370 nm (495 nm reference wavelength) on the Benchmark Plus microplate reader (Bio-Rad).

### 2.8. Cell Viability Assays

Cell viability was determined by the Alamar Blue assay (75 mg resazurin, 12.5 mg methylene blue, 164.5 mg potassium hexacyanoferrate (III), 211 mg potassium hexacyanoferrate (II) trihydrate in 500 mL of phosphate buffered saline). After 6 h incubation with Alamar Blue, the change of fluorescence was measured by Victor 3 multilabel Plate Reader (Perkin Elmer Australia, Melburne, Australia) at an excitation wavelength of 560 nm and an emission wavelength of 590 nm.

### 2.9. Western Blots

Protein was extracted from cell pellets using RIPA buffer (Sigma) with 10% protease inhibitor (Sigma, Sydney, Australia). Quantification of proteins was conducted using Pierce BCA protein assay kit (Thermo Scientific), according to manufacturer’s instructions. A total of 30–50 μg protein samples were loaded onto Criterion TGX 10% precast gel (Bio-Rad) or NuPAGE™ 4–12% Mini Protein Gel (Invitrogen, Life technologies) and transferred to a nitrocellulose membrane (Bio-Rad). The membrane was blocked in 5% skim milk in Tris-buffered saline with Tween-20 (20 mM Tris-HCl (pH 7.6), 137 mM NaCl, 0.1% Tween-20) for an hour before incubation at 4 °C overnight with the following primary antibodies: mouse anti-MYCN(1:2000, Santa-Cruz, Cat# sc-53993), rabbit anti-PA2G4 (1:2000, Atlas Antibodies, Cat# HPA016484), rabbit anti-PARP (1:1000, Cell Signalling Technologies, Danvers, MA, USA, Cat# 9542), rabbit anti-Caspase-3 (1:500, Cell Signalling Technologies, Cat# 9662), mouse anti-Bcl-2 (1:1000, Cell Signalling Technologies, Cat# 15071), rabbit anti-Bax (1:1000, Cell Signalling Technologies, Cat# 2772), mouse anti-Vinculin (1:2000, Sigma, Cat# V9131), mouse anti-GAPDH (1:2000, Santa-cruz, Cat# sc-365062), rabbit anti-β-actin (1:5000, Sigma, Cat# SAB2100037), rabbit anti-GATA2 (1:1000, Cell Signalling Technologies, Cat# 4595), mouse anti-Flag/DYKDDDDK tag (1:1000, Cell Signalling Technologies, Cat# 8146). Anti-mouse or anti-rabbit horseradish peroxidase secondary antibodies (1:5000, Life technologies, Cat# 31430, 31460) were added and incubated for 2 h at room temperature. Immunoblots were visualised by Clarity ECL reagent (Bio-Rad) and ChemiDoc MP Imaging System (Bio-Rad). Quantification of protein expression was measured by Image Lab software (Bio-Rad) and normalised to loading control.

### 2.10. siRNA and Plasmid DNA Transfections

For siRNA-mediated knockdown, 40 nM of Ambion Silencer Select GATA2 siRNAs (Thermo Fisher Scientific) (GATA2 siRNA#1: 5′-GGCUCGUUCCUGUUCAGAATT-3′, GATA2 siRNA #2: 5′-GGUACAGCUGUAUAUAAACTT-3′) (Cat# AM16704) were transfected with lipofectamine 2000 (Life Technologies), according to manufacturer’s instruction. Dharmacon on-target plus control siRNA (Cat# D-001810-10-20) was used as siControl. Cells were transfected between 24, 48, and 96 h, depending on the experimental requirements. For overexpression, 1 µg/mL of pCMV6-GATA2-Myc/DDK (Origene, Rockville, MD, USA) was transfected with pCMV6-Empty (Origene) as control. Lipofectamine 2000 (Life Technologies) was used, following manufacturer’s instruction.

### 2.11. Flow Cytometry

Neuroblastoma cell lines were seeded 24 h before treating with #5333 or #5338 for 48 h. The apoptotic and necrotic effects of the #5333 and #5338 treatment were determined by staining the cells with Annexin V PE/7-AAD using the Annexin V: PE Apoptosis Detection Kit (BD Biosciences, North Ryde, NSW, Australia), according to the manufacturer’s instruction. Stained cells were measured by flow cytometry using the BD LSRFortessa (BD Biosciences, NSW, Australia) and the data was analysed using the FlowJo software.

### 2.12. Methods for LFQ LC/MS

SK-N-BE(2)-C cells were incubated with #5333 or #5338 at IC50 concentration of the compound for 72 h. Cells were harvested and the pellets were lysed with RIPA buffer containing protease inhibitors (Sigma-Aldrich, St. Louis, MO, USA). Cell lysates were sent for LFQ/LC/MS analysis at the Bioanalytical Mass Spectrometry Facility, UNSW Sydney. Differentially expressed proteins where identified as those that had a *p*-value < 0.05 and a log2 fold-change (log2FC) of >1.5 for upregulated and <−1.5 for downregulated proteins. Differential protein expression analysis was performed using the DEP (v1.14.0) [22] package in R(v4.1.1). Variance stabilising transformation (vsn) method was used to normalise the protein expression data. Threshold of *p*-value < 0.05 and log2 fold-change value > |0.6| was used as a cut off to identify differentially expressed proteins. Functional enrichment analysis was performed using DAVID (v 6.8) [23].

### 2.13. Co-Immunoprecipitation Assay

SK-N-BE(2)-C cells were incubated with #5333 (53 µM) or #5338 (47 µM) at 2 × IC50 concentration of the compound for 1 h. The protein was harvested and the pellet was lysed in ice-cold BC100 buffer (20 mm Tris-HCl, pH 7.9, 100 mm NaCl, 10% glycerol, 0.2 mm EDTA, 0.2% Triton X-100, and freshly supplemented protease inhibitor) and incubated on ice for 30 min. After centrifugation at 12,000× *g* for 20 min at 4 °C, the supernatant was collected. For immunoblotting, 20–40 µg (calculated using the ThermoFischer Scientific BCA Assay kit) of whole protein lysates were resolved on either 10.5% or 10–14% Tris-HCl Criterion gels (Bio-Rad, South Granville, NSW, Australia). MYCN and GAPDH antibodies were purchased from Santa Cruz Biotechnology. A total of 750 µg of cell lysate was incubated with either MYCN or control mouse IgG antibody, then captured by Gammabind G-Sepharose beads (GE Healthcare). Bound proteins were resolved by SDS-PAGE. All primary antibodies were probed overnight at 4 °C and secondary antibody incubation was carried out at room temperature for 2–4 h. Immunoblots were visualised after incubation with Clarity™ Western ECL substrate and imaged with the ChemiDoc-Touch Imaging System (Bio-Rad, NSW, Australia).

### 2.14. Statistical Analysis

All statistical analysis was conducted by GraphPad Prism 9 software and differences were analysed using unpaired two-sided *t*-tests. All data were expressed as mean ± standard error of combined data from at least three independent biological replicates.

## 3. Results

### 3.1. Identification of Structural Analogues of WS6 with an Improved Therapeutic Window In Vitro

We have shown that the PA2G4 protein increased the half-life of the oncoprotein MYCN through direct binding, and competitive chemical inhibition of this bond by WS6 reduced tumorigenicity both in vitro and in vivo [11]. However, WS6 exhibited significant side-effects in murine models, resulting in a narrow therapeutic window when toxicities to malignant and non-malignant cells in vitro were compared. Here, we explored the structure–activity relationship (SAR) of WS6 analogue compounds with the aim to increase efficacy and reduce toxicity. We performed an initial in silico analogue screen on WS6, which resulted in 14 WS6 analogues (generation 1) with ≥80% chemical similarity to WS6 (Figure 1A,B and Appendix A). The in vitro cytotoxicity of these 14 compounds were measured by cell viability assays using Alamar Blue in MYCN amplified human neuroblastoma (SK-N-BE(2)-C and Kelly) cell lines compared to non-malignant (WI-38, MRC-5) myofibroblast cell lines (Figure 1B,C and Appendix A). This initial screen identified one compound #3567 (Figure 1B,C), which showed a larger therapeutic window of 2.7-fold compared with 2.1-fold for WS6 (Figure 1A–C).

We confirmed that #3567 interacted directly with recombinant PA2G4 by SPR with a K_D_ of 14.9 µM (Figure 1D). Protein expression of both PA2G4 and MYCN was significantly reduced in SK-N-BE(2)-C and Kelly cells (Appendix A) with treatment by #3567 (also called SVI-3567). Seeking a further reduction of toxicity and improvements in physicochemical properties, we conducted a second in silico analogue screen using the structure of #3567. A further 33 compounds were purchased and assayed (Appendix A). Of these compounds, 5 showed dose-response direct binding to recombinant PA2G4 via SPR, with compounds #5333 and #5338 showing a K_D_ of 10.8 µM and 13.5 µM, respectively (Figure 1D). Differential scanning fluorimetry (DSF) showed that #5333 and #5338 increased the melting temperature of PA2G4, further indicating binding avidity of the compounds for the PA2G4 protein (Appendix A). Cellular assays showed that these compounds inhibited MYCN amplified human neuroblastoma cell lines with an IC50 of 26.5 µM for #5333 and 23.3 µM for #5338 in SK-N-BE(2)-C cells, and 28.6 µM for #5333 and 30 µM for #5338 in Kelly cells (Figure 1B). Importantly, these compounds displayed the largest therapeutic window of 3.6 and 3.8, respectively (Figure 1B,C), compared to 2.1 for WS6.

To further explore the SAR and aid in the design of future compounds, the analogues were computationally docked against the crystal structure of PA2G4 (Figure 1E). Two docking algorithms were utilised, with the active compounds docking into the PA2G4 crystal structure in the same orientation for both programs. This docking orientation displays three putative hydrogen bonds; two of these are with Arg272 via the pyrimidine and the nitrogen, the other is between the carbonyl oxygen and the backbone of Ile74. The R1 position stacks above Ala54. Our SAR studies showed 2,3 dimethyl phenol was beneficial over either an unsubstituted phenol or a 4-ethly phenol. The 2,3 dimethyl allows for the hydrophobic group to be buried against the carbon side chain of Glu50 and Arg281, as opposed to the 4-ethyl substituent, which we propose would be exposed from the protein binding pocket. The R2 substituent of #3567 and #5333 contains a putative interaction between the 3-CF3 or 3-Br group and the backbone carbonyl oxygen of Arg271. Although #5338 does not contain an electronegative group in this position, the additional carbon atoms of the R2 substituent are postulated to interact with Met276 and the side chain carbons of Lys71. It is also worth noting that Lys71 is adjacent to a flexible loop region; thus, this may move as the compound binds to increase interactions with #5338.

Importantly, all compounds had significantly improved physicochemical properties when compared to the parent compound WS6 (Figure 1F). Specifically, clogP was reduced from 7.3 for WS6 to 4.9, 4.8, and 5.3, respectively, for #3567, #5333, #5338. Although still higher than preferred, the modelling has suggested areas where heteroatoms could be incorporated to reduce this significantly. For example, the introduction of a heteroatom to the R1 substituent could allow for a hydrogen bond to Glu50 or Arg271. Notably, although the introduction of heteroatoms would increase the molecular weight, the analogues had significantly reduced molecular weight (150–200 Da lower) when compared to WS6. Furthermore, numerous studies have shown that protein-protein interaction inhibitors do tend to be larger and suggest that clogP is a better measure of the potential for binding promiscuity.

### 3.2. WS6 Analogues Inhibit the Malignant Phenotype of Neuroblastoma Cells In Vitro

We treated human neuroblastoma tumour cell lines with high MYCN and PA2G4 expression (Kelly and SK-N-BE(2)-C), using increasing concentrations of #5333 or #5338, and measured the ability of cells to proliferate and form colonies in vitro over 2 weeks. Treatment with #5333 or #5338 decreased both the number and size of the colonies in Kelly (Figure 2A, Appendix A) and SK-N-BE(2)-C (Figure 2B, Appendix A) cells. Next, we evaluated cell replication using the BrdU uptake assay in response to 30 μM drug treatment for 48 h (Figure 2C). Neuroblastoma cell proliferation was significantly reduced by both #5333 and #5338 in neuroblastoma cells (Figure 2C). Collectively, these results confirm significant in vitro anti-cancer effects of #5333 and #5388 at concentrations of 10–30 μM.

To determine if the drug effects led to apoptosis, we stained cells with annexin-V/7-AAD, a marker of early and late apoptotic events (Figure 2D). Analogues #5333 and #5338 caused similar levels of apoptosis in neuroblastoma cells. To confirm the apoptotic status of the cells, protein expression of key apoptotic markers PARP/Cleaved PARP, Caspase 3/Cleaved caspase 3, BCL2, and Bax was measured (Figure 2E). The expression of cleaved PARP and cleaved Caspase 3 was increased in response to #5333 and #5338 treatment, compared to WS6 treatment, at an IC50 concentration (0.65 µM for Kelly and 0.75 µM for SK-N-BE(2)-C), but without changes in Bcl-2 or Bax (Figure 2E). These data indicate that #5333 and #5338 were more effective than WS6 in activating apoptotic pathways leading to cell death.

### 3.3. The Cytotoxicity of WS6 Analogues Is Partially Dependent on PA2G4 and MYCN Protein Expression

Next, we examined whether #5333 and #5338 treatment for 48 h affected the levels of WS6 target proteins, PA2G4, and MYCN in neuroblastoma cells [11]. Western blot analysis showed that treatment of Kelly and SK-N-BE(2)-C, with both compounds, significantly reduced MYCN levels at concentrations of 10–30 μM. Reductions in PA2G4 levels, after drug treatment, were seen more consistently at 30 μM concentrations, with the exception of #5333 treatment of SK-N-BE(2)-C cells (Figure 3A). To determine the specificity of #5333 and #5338 for MYCN, we used two different doxycycline (Dox)-inducible neuroblastoma cell systems for repression of MYCN expression: shMYCN SK-N-BE(2)-C (MYCN-amplified) and shMYCN Kelly (MYCN-amplified) cell lines. Cell viability was measured following #5333 and #5338 treatments for 48–72 h. Reduction of MYCN expression after Dox treatment resulted in a significant increase in IC50 for both compounds compared with DMSO controls and WS6 treatment in shMYCN SK-N-BE(2)-C and shMYCN Kelly cells (Figure 3B–G and Appendix A). In addition, the specificity and efficacy of #5333 and #5338 were also tested in MYCN non-amplified SHEP MYCN3 cell lines with Dox-inducible MYCN induction. Cell viability was measured following #5333, #5338, or WS6 treatments for 72 h. Induction of MYCN expression after Dox treatment resulted in a significant decrease in IC50 for both #5333 and #5338 compounds compared with DMSO treated controls and WS6 treated SHEP MYCN3 cells (Appendix A–J). This data suggests that the cytotoxicity of #5333 and #5338, for neuroblastoma cells, partially depends on MYCN.

Since WS6 disrupts the binding of MYCN to PA2G4, with a consequent effect on MYCN stability [11], we hypothesised that the two WS6 analogues bound these two target proteins in a similar manner. Next, we incubated SK-N-BE(2)-C cells with twice the IC50 concentrations of the compounds for 1 h and performed a Co-IP, which showed that the two WS6 analogues reduced direct binding of PA2G4 to MYCN (Figure 3H) in a similar manner to the parental compound, WS6 [11].

### 3.4. Quantitative Proteomics Reveals Potential Biomarkers of WS6 Analogue Anti-Cancer Actions

To further understand the mechanism of action and off target effects of the WS6 analogues, we used label-free quantitative mass spectrometry of differential proteome expression in response to drug treatment. SK-N-BE(2)-C cells were treated with IC50 concentrations of either #5333 (26.5 µM), #5338 (23.3 µM), or DMSO solvent control for 72 h, and whole cell lysates were analysed by LC/MS, as previously described [24]. A total of 4255 unique proteins were captured across the three proteomes, of which 258 proteins were differentially expressed after treatment with #5333, and 1371 proteins treated with #5338, when compared to DMSO-treated cells (Figure 4A,B and Appendix A).

The significantly differentially expressed proteins for #5333 and #5338 were analysed using the Gene Ontology (GO) database to determine biological processes downregulated (Figure 4C,D) or upregulated (Appendix A), due to the activity of the compounds. Differential protein expression analysis of the shared targets between the two compounds showed 99 upregulated proteins (Appendix A) and 50 downregulated proteins (Appendix A) between both compounds. Amongst the shared common pathways, cell proliferation and regulation of mitotic cell cycle pathways were significantly downregulated by both compounds (Figure 4C,D). Since MYCN and MYC share many transcriptional target genes and mechanisms, we compared the differentially expressed proteins following WS6 analogue treatment with 1470 MYC target proteins, as described previously [25]. A significant change in protein expression was observed in MYC target genes after treatment with both compounds (Figure 4E) and is summarised in a table (Appendix A). This identified GATA2 as the most differentially downregulated protein, with a fold change of −4.2 for #5333 and −5.5 for #5338 (Figure 4E and Appendix A). GATA2 has been identified as a component of the transcriptional regulatory circuits involving super enhancers in neuroblastoma cells, which also includes GATA3, PHOX2A, PHOX2B, and HAND2 [26]. GATA2 has been shown to function as an oncoprotein in many other cancers [27]. Therefore, we sought to validate GATA2 as a target regulated by our two compounds.

### 3.5. GATA2 as a Potential Downstream Target Mediating the Anti-Cancer Effects of #5333 and #5338

Western blot analysis of SK-N-BE(2)-C and Kelly cells showed that, at IC50 concentrations, both compounds significantly reduced GATA2 and MYCN levels at 72 h (Figure 5A). Next, we overexpressed GATA2 plasmid DNA in the two neuroblastoma cell lines and measured cell viability in response to #5333 and #5338 treatment but were unable to show a reduced drug effect on cell viability in the presence of excess GATA2 (Appendix A). However, when we transiently transfected SK-N-BE(2)-C and Kelly cells with two GATA2-specific siRNAs for 96 h, in comparison with siRNA controls, the siRNA knock-down of GATA2 caused a further decrease in cell viability after 72 h in #5333 and #5338 treated cells, suggesting that GATA2 may be regulated by MYCN/PA2G4 and the further reduction of the residual GATA2 by siRNAs, in the neuroblastoma cells treated with the compounds, suppressed the cell viability additively (Figure 5B–D and Appendix A).

## 4. Discussion

We have previously reported that WS6 inhibits the MYCN-PA2G4 protein-protein interface and markedly reduces MYCN protein stability and neuroblastoma tumorigenesis. Here, we identified two analogues of WS6 with 80% structural similarity and a 70-fold lower toxicity for normal human myofibroblasts than WS6. WS6 analogues also inhibited the malignant neuroblastoma cell phenotype in vitro; their cytotoxicity was partially dependent on PA2G4 and MYCN protein expression. We also found that a transcription factor and MYC target, GATA2, is a downstream target for the anti-cancer effects of the WS6 analogues. Our data provides a basis for the future design of competitive inhibitors of the PA2G4-MYCN protein interface.

In recent years, many groups attempted to target MYCN oncogenic signalling at various stages of its formation and function. Inhibitors of MYCN downstream oncogenic signalling pathways have also been developed [28]. However, inhibition of MYCN protein binding partners (MYCN-BPs) is emerging as an efficient strategy for targeting MYCN oncogenic signalling in the treatment of MYCN-driven cancers [6,7,10,11]. A variety of functions of the MYCN gene product are attributable to protein-protein interactions (PPIs). Extensive evidence, from our lab and others, have shown that MYCN-BPs can markedly increase MYCN protein stability to drive tumour formation [9,11,29,30]. In neuroblastoma cells, PA2G4 is a transactivation target for MYCN. PA2G4 directly binds to and stabilises MYCN by preventing its ubiquitin-mediated proteolysis [11]. Inhibition of PA2G4 and MYCN binding by WS6 can reduce levels of both proteins and neuroblastoma tumorigenicity in vitro and in vivo. In this study, the two WS6 analogues displayed a much larger therapeutic window compared to WS6. Nonetheless, potential off-target effects of the WS6 analogues need to be investigated. Additionally, the half maximal inhibitory concentration (IC50) of two WS6 analogous in treated neuroblastoma cells was relatively high; therefore, chemical modification to synthesise more potent WS6 analogues is required to improve the effectiveness on neuroblastoma cells in future study. As the 2,3 dimethyl phenoxy ring at R^1^ position was important for cytotoxicity (Figure 1B), we plan to keep this group constant at R^1^ position and explore the R^2^ position to enhance activity. First, we would explore the effect of bioisosteric equivalents of the trifluoromethyl (CF_3_) group, such as iodide (I), isopropyl (iPr), and tert-butyl (tBu), at the meta position of the phenyl ring in R^2^ position. Second, we would explore the effect of the carbon chain length between amino (NH) group and the phenyl ring by a systematic increase of carbon linkage from 0–5 carbons for compound #5338. Third, we would explore the bioisosteric replacements of phenyl ring with thiophene, pyridyl rings. Fourth, we would synthesise pure enantiomers of #5338 to study the effect of spatial orientation of the methyl group on activity. In this study, SAR studies revealed that 2,3 dimethyl substitution on the phenoxy ring at R1 position was important for cytotoxicity of #3567 against neuroblastoma cell lines. The compound without this dimethyl substitution, #3561, was not cytotoxic towards neuroblastoma cell lines, SK-N-BE(2)C, and Kelly cells (IC50 > 100 µM). Furthermore, investigating the effect of trifluoromethyl (CF3) group at different positions of phenyl ring at the R2 position revealed that the CF3 group is required at meta position to the amino-methyl group, as #3567 displayed a 2.7-fold therapeutic window against normal cells. Moving the CF3 to the para-position of amino-methyl group decreased the therapeutic window to 1.4-fold and changing the CF3 to ortho-position of amino-methyl group resulted in complete loss of cytotoxicity (#3566, IC50 > 100 µM)). Replacing the CF3 group of #3567 with a bromine atom resulted in an increase in the therapeutic window to 3.6-fold. The series of compounds reduced the cytotoxic effect compared to the parent compound WS6, although there was an increase in the therapeutic window from 2.1- to 3.8-fold.

GATA2 is a transcription factor that plays an essential role in gene regulation during vascular development and hematopoietic differentiation. A study found that haploinsufficiency of GATA2 underlies primary lymphedema and predisposes to acute myeloid leukaemia [31]. GATA2 mutation is also one of the causes of B and NK lymphoid deficiency, which could contribute to leukemic transformation [32]. Importantly, GATA2 regulates the metastatic potential of prostate cancer cells in the early stages of disease. GATA2 has also been shown to increase the metastatic potential of androgen responsive LNCaP cells. Thus, targeting GATA2 is an attractive therapeutic strategy that might improve the clinical outcome of patients with prostate cancer [27,33]. A study in neuroblastoma suggests that GATA2 plays a role in ATRA-induced neuronal differentiation in neuroblastoma SH-SY5Y cells [34]. GATA2 may also play critical roles in maintaining adrenergic features in poorly differentiated tumours [35]. However, the functional role of GATA2 in neuroblastoma has not been fully studied. In this study, we found GATA2 was the most differentially downregulated protein after the treatment of #5333 and #5338, identified by a label-free LC/MS study, and the downregulation of GATA2 protein was confirmed by immunoblotting. GATA2 repression by siRNA increased the sensitivity of neuroblastoma cells to the two compounds, suggesting that GATA2 could be the downstream target in mediating the anti-cancer effects of #5333 and #5338. However, further studies are needed to determine the molecular mechanism for how GATA2 mediates the anti-cancer effects of these compounds.

## 5. Conclusions

Due to the ever-increasing knowledge regarding MYC biology and function, and continuous developments in drug screening methods, new MYC-targeting therapeutic opportunities constantly arise. The most efficient MYCN inhibition methods will likely be a combination of targeting methods, involving targeting more than one pathway. Many MYC inhibitors, including BET, CDK, and Aurora A inhibitors [36,37,38], have been limited by toxicities and poor therapeutic windows in clinical trials. Obtaining a good therapeutic window is a particularly important consideration in MYC inhibition, as MYC plays a crucial role in both physiological and oncogenic pathways. Using combination therapy of PA2G4 and MYCN inhibitors, with side effect profiles, may allow us to achieve an optimal therapeutic window whilst minimising the likelihood of drug resistance.

## Figures and Tables

**Figure 1 cancers-15-01822-f001:**
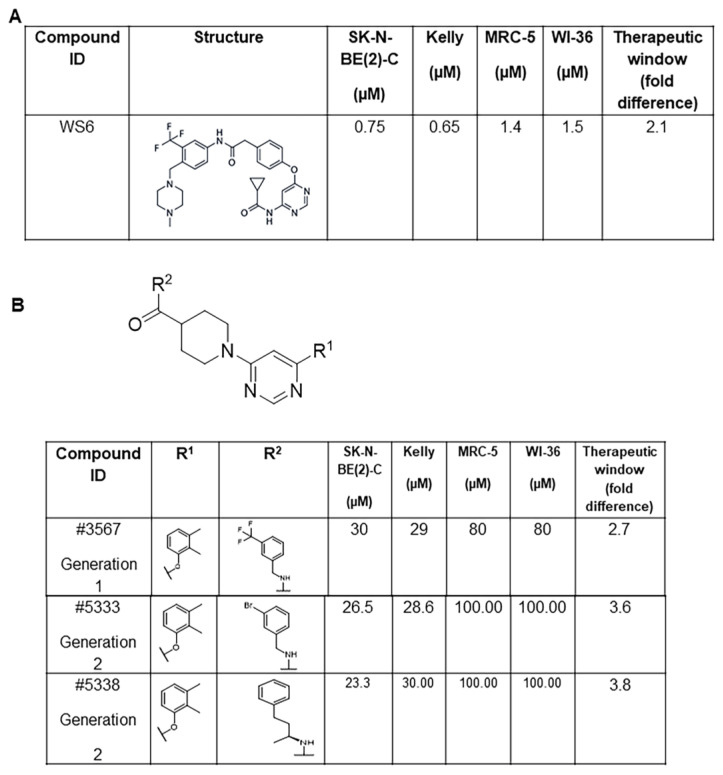
Identification of selective chemical inhibitors of PA2G4-MYCN oncogenic interaction through SAR studies. (**A**) WS6 structure and cytotoxicity in neuroblastoma. (**B**) Core structure of Generation 1 and 2 compounds. SAR comparisons of R1 and R2 substitutes of WS6 hit analogues. (**C**) Representative IC50 plots for WS6, #3567, #5333, and #5338 comparing neuroblastoma cells (Kelly, SK-N-BE(2)-C) to non-malignant myofibroblasts (WI-38, MRC-5). (**D**) Representative Surface Plasmon Resonance curves showing the concentration-dependent binding of #3567, #5333, or #5338 to PA2G4. (**E**) Docking orientation of SVI3567 (green), SVI-5333 (cyan), and SVI-5338 (magenta) in PA2G4 (white). Shown in sticks are the amino acids proposed to interact with the compounds. Specifically, there are hydrogen bonds to the main-chain atoms of Arg272 and Ile74 (shown as yellow dashed lines), and hydrophobic interacts with Ala54, Glu50, Arg281, Met276, and Lys71. Furthermore, there is a putative interaction between the 3-CF3 or 3-Br group of #3567 and #5333 and the backbone carbonyl oxygen of Arg271. (**F**) Summary of the physicochemical properties of the top WS6 analogues. Molecular weight (MW), partition co-efficient (clogP), hydrogen bond acceptors (HBA), hydrogen bond donors (HBD), polar surface area (PSA), and rotatable bonds (RB) were all calculated using Chemaxon Software and compared to the average (median) of all protein-protein inhibitors on the market. Median value of protein-protein interaction inhibitors approved for use [14].

**Figure 2 cancers-15-01822-f002:**
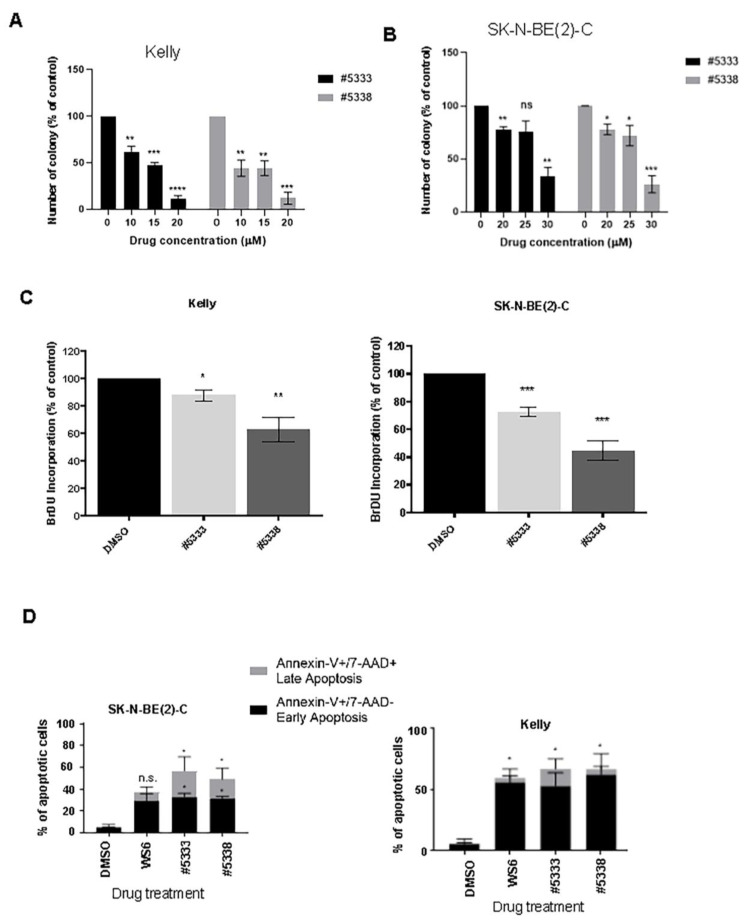
Compounds #5333 and #5338 inhibit the malignant phenotype of neuroblastoma cells in vitro. Colony formation assays measuring colony numbers at 2 weeks timepoint in (**A**) Kelly and (**B**) SK-N-BE(2)-C neuroblastoma cell lines. (**C**) BrDu assays measuring proliferation at compound IC50 concentrations for Kelly and SK-N-BE(2)-C cells. (**D**) Flow cytometry of apoptosis profiles of cells treated with #5333 (7.5 µM) and #5338 (6 µM), compared with untreated DMSO control treatment, using Annexin-V & 7-AAD stains. Bar charts depict cells in early or late apoptotic states as the percentage of total apoptotic cells. (**E**) Western blot analysis of changes in apoptotic effector proteins PARP, Caspase-3, Bcl-2, and Bax in response to #5333 or #5338 treatment. Values are mean ± SEM (n = 3). Asterisk (*) refers to comparison of treated groups to DMSO control groups. * *p* < 0.05, ** *p* < 0.01, *** *p* < 0.001, **** *p* < 0.0001, ns: non-significant. Statistical significance was determined using Student’s *t*-test. The uncropped blots are shown in Appendix A of original Western Blot figures.

**Figure 3 cancers-15-01822-f003:**
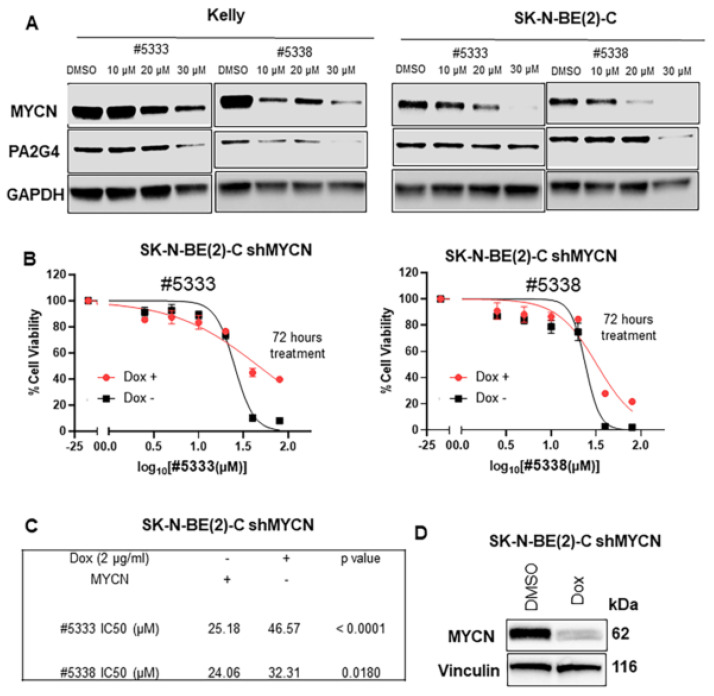
Compounds #5333 and #5338 functionally interact with MYCN and PA2G4 in neuroblastoma cells. (**A**) Western blot showing a decrease in protein expression due to #5333 or #5338 drug treatment. (**B**) Alamar Blue assays comparing cell viability of SK-N-BE(2)-C after Doxycycline (Dox)-inducible MYCN knockdown, followed by 72 h treatment with #5333 or #5338. (**C**) IC50s of Dox-inducible SK-N-BE(2)-C shMYCN cells treated with or without Dox and compound #5333 or #5338. (**D**) Western blot measuring MYCN expression in Dox-inducible SK-N-BE(2)-C shMYCN cells treated with DMSO or 2 μg/mL of Dox for 96 h. (**E**) Alamar Blue assays in Kelly shMYCN comparing cell viability of Dox-inducible MYCN knockdown, followed by 72 h of #5333 or #5338 treatment. (**F**) IC50 values after 72 h of #5333 or #5338 treatment of the Dox-inducible Kelly shMYCN cell line. (**G**) Western blot measuring MYCN protein expression in Dox-inducible Kelly shMYCN cells treated with DMSO or 2 μg/mL of Dox for 96 h. (**H**) A MYCN Co-IP analysis using protein from SK-N-BE(2) cells treated with compounds #5333 or #5338 treatment, and immunoblotted with antibodies recognising MYCN, PA2G4, GAPDH, or Light Chain IgG. The uncropped blots are shown in Appendix A of original Western Blot figures.

**Figure 4 cancers-15-01822-f004:**
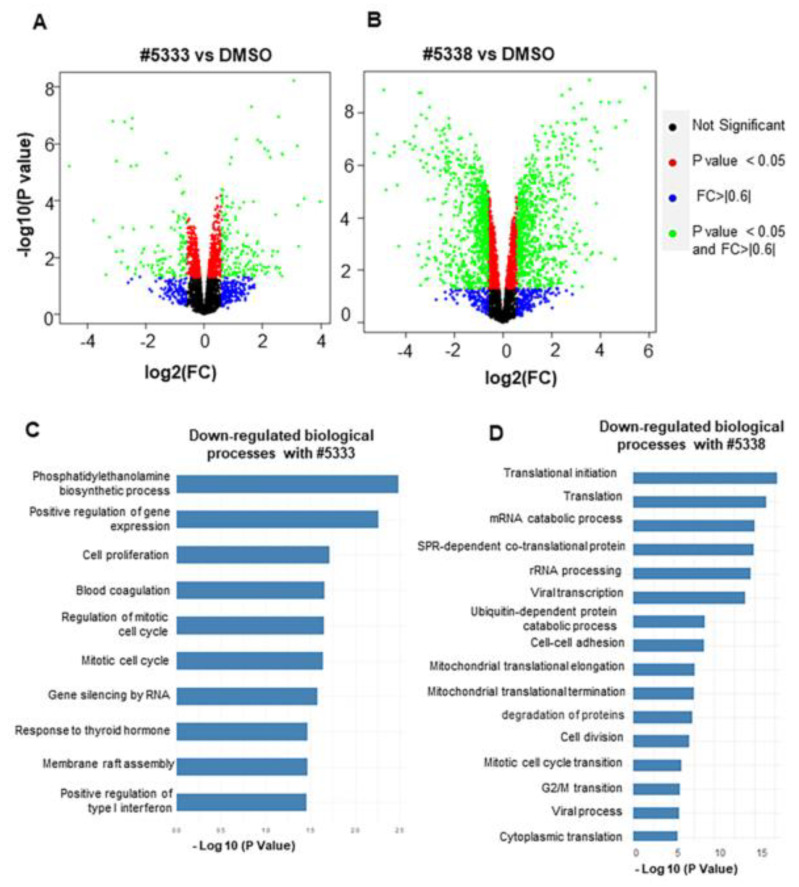
Label-free quantitative mass spectrometry of differentially expressed proteins from neuroblastoma cells following treatment with compounds #5333 or #5338. (**A**,**B**) SK-N-BE(2)-C cells were treated with 2 x the IC50 concentrations of #5333, #5338, or DMSO control and were incubated for 72 h, followed by total protein lysate collection and analysis using LCMS/MS. Three biological replicates were processed and analysed for each group. Volcano plot of the differentially expressed proteins of (**A**) #5333- and (**B**) #5338-treated cells compared to DMSO. (**C**,**D**) Downregulated Gene Ontology (GO) biological processes −log10 *p* value bar plots for (**C**) #5333- and (**D**) #5338-treated cells. (**E**) Known MYC target genes modified by #5333 or #5338 treatment.

**Figure 5 cancers-15-01822-f005:**
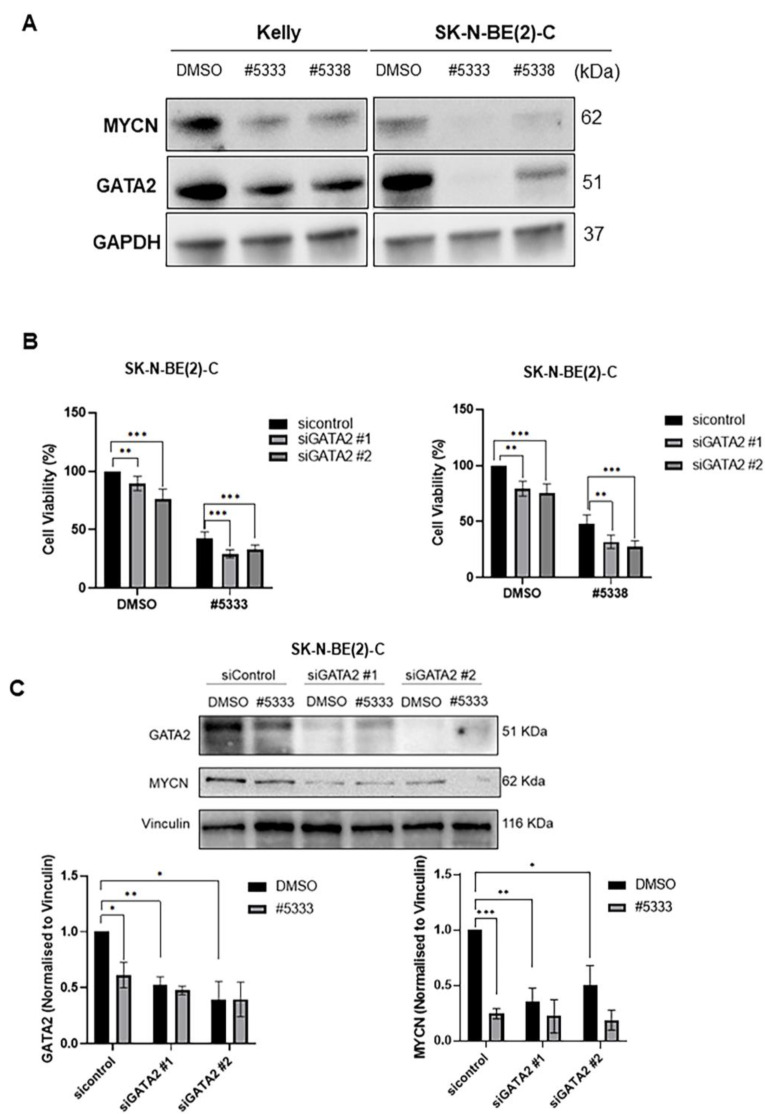
GATA2 as a downstream target for anti-cancer effects of #5333 and #5338. (**A**) Representative Western blot analysis for GATA2 and MYCN expression in Kelly and SK-N-BE(2)-C cells treated with #5333 or #5338 at IC50 concentrations for 72 h. (**B**) SK-N-BE(2)-C cell viability was measured after transfected with siRNA control and two GATA2 siRNAs (siRNA#1 and #2) for 96 h, and treated with #5333 (26.5 µM) or #5338 (23.3 µM) for 72 h using Alamar Blue assay. (**C**,**D**) Immunoblot analysis of GATA2, MYCN, and control Vinculin expression in SK-N-BE(2)-C after transfection with siRNA control or two GATA2 siRNAs (siRNA#1 and #2) for 96 h, and treatment with #5333 (**C**) or #5338 (**D**) for 72 h. The uncropped blots are shown in Appendix A of original Western Blot figures. “*” is *p* value < 0.05, “**” is *p* value < 0.01 and “***” is *p* value < 0.001 when compared to DMSO treated control.

## Data Availability

No new data were created.

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
