# Peer review of "Inhibitors of the Oncogenic PA2G4-MYCN Protein-Protein Interface"

_cancers, 2023, doi:10.3390/cancers15061822_

Round 1

Reviewer 1 Report (Previous Reviewer 2)

Authors responded to my suggestions and part of them are acceptable. I suggested the following points to improve this paper to publish on Cancers.

Point 1.

Authors indicated that the half maximal inhibitory concentration (IC50) of two WS6 analogous in treated neuroblastoma cells was relatively high and chemical modification to synthesize more potent WS6 analogues is required to improve the effectiveness on neuroblastoma cells in the future study. If authors are really thinking about the chemical modification to synthesize more potent WS6 analogues, they need to mention the specific plan to modify the WS6 analogous.

Point 2.

Authors indicated that “Photos of colonies were taken by ChemiDoc MP Imaging System (Bio-rad) and colonies were counted using Image J software”. I recommend them to mention about measurement the size of the colonies in 2.6. Colony formation assays.

Point 3.

I still think that IC50 of 26.5 μM for #5333 and 23.3 μM for #5338 in SK-N-BE(2)-C cells, and 28.6 μM for #5333 and 30 μM for #5338 in Kelly cells are too high to do apoptosis induction experiments. They need to show the apoptosis induction data using less than 10 micro M.

Point 4.

OK.

Point 5.

I accepted the reply.

Point 6.

I accepted the reply.

Point 7.

I accepted the reply.

Point 8.

I still believe that NB xenograft experiments will be required to confirm the authors’ argument described the effectiveness and the improved toxicity of the compounds. That data will improve the quality of the present paper.

Author Response

Reviewer 2 Report (Previous Reviewer 3)

The manuscript has been significantly improved and now warrants publication in cancers

Author Response

Thank you very much for your comments. 

This manuscript is a resubmission of an earlier submission. The following is a list of the peer review reports and author responses from that submission.

Round 1

Reviewer 1 Report

The study reported by et al., about Inhibitors of the oncogenic PA2G4-MYCN protein-protein interface. In brief, the authors characterized and evaluated WS6 analogous as inhibitors of MYCN and PA2G4 oncogenic functions in neuroblastoma. The overall manuscript is well written, the study will certainly expand our knowledge of drug discovery toward MYCN-driven cancers

There are no major points for this work. with incorporation of following minor changes, I would enthusiastically endorse publication of this article.

1.    There is a possibility to polish the English throughout the manuscript, authors may concise the statements /avoid lengthy statements.

2.    Resolution of the figures need to be improved, especially fonts of labels can be converted to more legible version

Please check the attached PDF for comments which need to be addressed/

Reviewer 2 Report

Massudi M et al identified two analogues of WS6 with 80% structural similarity and a 70-fold lower toxicity for normal human myofibroblasts than WS6. The analogues also inhibited the malignant NB cell phenotype in vitro, their cytotoxicity was partially dependent on PA2G4 and MYCN protein expression. Authors also found that a transcription factor and MYC target, GATA2, is a down-stream target for the anti-cancer effects of the WS6 analogues. Their data provides a basis for future design of competitive inhibitors of the PA2G4-MYCN protein interface.

PA2G4 seems to be an interesting target for the MYCN-amplified NB treatments and the screening of the PA2G4 inhibitors will be an important project. However, several points in this paper were unclear and the additional experiments will be required for acceptance in the journal.

Comment 1

>Cellular assays showed 307 that these compounds inhibited MYCN amplified human neuroblastoma cell lines with 308

>an IC50 of 26.5 μM for #5333 and 23.3 μM for #5338 in SK-N-BE(2)-C cells, and 28.6 μM 309

>for #5333 and 30 μM for #5338 in Kelly cells (Fig.1B).

IC50 of these compounds are too high. This reviewer thinks that the clinical trials using these compounds will not be possible because of the IC50. Authors need to try modification of the compounds to improve the effectiveness on NB cells.

Comment 2

>weeks. Treatment with #5333 or #5338 decreased both the number and size of the colonies 367

>in Kelly (Fig. 2A; Supplementary Fig. S2A) and SK-N-BE(2)-C (Fig. 2B, Supplementary 368

Why 2 of Kelly panels and 2 of SK-N-BE(2)-C panels were here (Fig.2A)? Time course experiments will be more informative?

Comment 3

>These data indicate 381

>that #5333 and #5338 were more effective than WS6 in activating apoptotic pathways lead-382

>ing to cell death. 383

Again, the concentration of the #5333 and #5338 to induce apoptosis were very high.

Comment 4

In Figure 3 C and F, also in Supple Figure 3 B and D, #5338 is missing.

Comment 5

>This data suggests that MYCN is necessary for the cytotoxicity 413

>of #5333 and #5338 for neuroblastoma cells. 414

Although MYCN reduction increased the IC50 of the compounds, the effects were not so obvious, suggesting that the cytotoxicity of #5333 and #5338 for NB cells partially depends on MYCN.

Comment 6

>We next incubated SK-N-BE(2)-C cells with twice the 417

>IC50 concentrations of the compounds for 1 hour and performed a Co-IP, which showed 418

>that the two WS6 analogues reduced direct binding of PA2G4 to MYCN (Fig. 3H) in a 419

>similar manner to the parental compound, WS6 [11]. 420

The heavy chain signals were too strong to evaluate the precipitated MYCN amounts in Figure 3H. Authors need to use the appropriate 2nd Ab kits to reduce the heavy chain signals.

Comment 7

>However, when we transiently transfected SK-N-BE(2)-C 487

>cells with two GATA2-specific siRNAs for 96 hours, in comparison with siRNA controls, 488

>the siRNA knock-down of GATA2 caused a further decrease in cell viability after 72 hours 489

>in #5333 and #5338 treated cells, suggesting the sensitivities of the two compounds are 490

>reduced in neuroblastoma cells following GATA2 repression (Fig. 5B-D). 491

Figure 5 experiments suggested that GATA2 may be a gene regulated by MYCN/PA2G4 and the further reduction of the residual GATA2 by GATA2 siRNAs in the NB cells treated with the compounds suppressed the cell viability additively.

Comment 8

NB xenograft experiments will be required to confirm the authors’ argument described the effectiveness and the improved toxicity of the compounds.

Reviewer 3 Report

MYCN is the well-known oncogene in neuroblastoma, a pediatric cancer. Myc protein family including MYCN is prevalent in different cancer types but it has been difficult to design chemical drugs due to a lack of rigid structure by its intrinsically disordered region. The authors previously found PA2G4 as a directly interacting protein with MYCN and showed that disrupting their binding by a small chemical WS6 resulted in the growth suppression of MYCN-driven neuroblastoma in vitro and in vivo (Ref. 11). However, WS6 showed a limited drug efficacy because of its narrow therapeutic window, in other words, it showed non-negligible side effects. In this manuscript, Hassina Massaudi, Jie-Si Luo, Jessica K. Holien, et al, performed in silico screening with structural information to further refine its efficacy and reached to two analogues (named #5333 and #5338). They used two MYCN-amplified neuroblastoma cell lines (SK-N-BE(2)-C and Kelly) and two non-malignant fibroblast cell lines (WI-38 and MRC-5), and performed a series of in vitro assessments. Proteomics analysis identified downstream targets of WS6 analogues, and they selected GATA2 as a candidate gene for anti-cancer effects by them. The manuscript is original and valuable for neuroblastoma research community to some extent. However, it still has several concerns as summarized below and thus is not recommendable for publication in Cancers in its present form.

-        In Figure 3, the authors used inducible shRNA to knockdown MYCN to assess whether #5333 and #5338 specifically affected neuroblastoma cell survival. However, MYCN knockdown itself should lead to significant cell death in MYCN-amplified neuroblastoma cells and thus this experimental setting is not suitable to properly assess the specific effects of these drugs. Therefore, assessment of drug efficacy in MYCN non-amplified cell lines with MYCN induction is preferred to conclude.

-        Main point of this manuscript is to compare the efficacy of original WS6 to two analogues. In some experiments in Figure 3, the authors only used #5333 and #5338 without WS6 and thus superiority was not able to properly assess.

-        In figure 5B, the authors should include the result of the same experiment using Kelly.

-        This reviewer is not satisfied with the experiments regarding GATA2. Although the authors found that GATA2 was a downstream protein by WS6 analogues, GATA2 contribution is very limited in this context according to the results shown in Figure 5. I was wondering if the authors assessed other targets as well.

-        Significant off-target effects were problem of WS6, and the authors wanted to find new drugs having better therapeutic window with lower side-effect. The authors only used two non-malignant fibroblasts in this study. Is there any evidence from in vivo experiments that two WS6 analogues show less side-effects compared to original WS6?

Minor points

-        What is the reason to use two non-malignant fibroblasts MRC-5 and WI-38 in this study? It might be better to use non-malignant cells which express PA2G4. Is PA2G4 expressed in these cells?

-        Line 294-296: I noticed that #3557 showed the best therapeutic window with 3.21-fold higher than others in the initial screening. Readers may be wondering why the authors selected #3567 rather than #3557.

-        In Figure 3H, blotting for MYCN is not clearly seeable due to heavy chain signals. If the authors used mouse antibody for IP, it is recommended to use rabbit antibody (for example, #51705 from Cell Signaling Technology) for western blotting.

-        Line 82: I was a bit confused with the word “cytopathic effects” that generally mean structural changes by virus infection, and thus may be better to use other words in this context.

-        Line 287-289: In the sentence, “Here explored the structure-activity relationship...”, the subject is missing.

Reviewer 4 Report

The authors present an interesting study of new small size inhibitors, consisting of WS6 analogues, that interfere with the interaction between the proteins PA2G4 and MYCN. Because the former protein is a cofactor of MYCN, disruption of this protein interaction has a clear antioncogenic potential. The study builds on previous reports by the authors on the druggability of the PA2G4-MYCN interaction and the analysis of small molecule modulators of protein-protein interactions. Overall, the manuscript is well written and the study supported by appropriate methods and statistics analyses. However, there are some important issues that must be addressed to improve the quality of the report. These are described as follows:

Figure 3 shows a decrease in MYCN and PA2G4 protein expression levels following treatment with the compounds #5333 and #5338. In the introduction section it was mentioned that binding of PA2G4 to MYCN results in the stabilisation of the latter protein, preventing its degradation by the Ubiquitin Proteasome System, which explains the reason as to why MYCN protein levels drop in neuroblatoma cells treated with compounds #5333 and #5338. However, it far less clear the reason for the decrease of PA2G4 protein levels upon treatment with the aforementioned WS6 analogues. Such compounds are expected to bind PA2G4, resulting in a stabilising effect. Indeed, DSF data support this view as it shows a shift of the Tm of PA2G4 in the presence of these small compounds (Suppl Fig S1D). The authors should discuss these apparently contradictory results regarding PA2G4 stability/degradation upon treatment with the new WS6 analogues. The authors mentioned that MYCN transcriptionally regulates multiple target genes. Could PA2G4 be one of them? 

Conclusions. In the sentence "Using combination therapy of 573 PA2G4 and MYCN inhibitors with complementary side effect profiles..." please clarify what you mean by complementary side effect profiles.

There are some minor issues that also ought to be addressed. These are listed as follows: 

Materials and Methods, section 2.3 shows a mix of font styles. Please amend. 

-Figure 1, Panel C. In the graphs showing WS6 and compound 3567, the symbol to denote micromolar is not completely legible, probably because it is shown in bold. Make sure the style is the same as that shown in the graphs of compounds 5333 and 5338. 

-Figure 1, Panel D. In addition to these graphs presenting the IC50 plots, please include a figure (this could be presented as supplementary material) showing the sensograms of compounds at different concentrations versus time. Presenting the data in this way will make easier to visualise the association/ dissociation events for each compound at different concentrations and compare the relative binding affinity of the molecules. 

Figure 1, Panel E. compounds #3567, #5333 and #5338 are also referred to as SVI-3567, SVI-5333 and SVI-5338. The dual nomenclature is also used in the discussion of results section. This is unnecessary and can cause some confusion. Therefore, I recommend the authors to stick to one nomenclature when referring to these compounds throughout the manuscript.

Figure 2, X-axis legends are shown in different font size. Please present Panels C and D as graphs of similar size. This is particularly important for Panel D, as the legends are not totally legible. 

Figure 3, Panels C and D. Can some of the biological processes listed here be simplified? I found it confusing to refer to "Regulation of mitotic cell cycle" and "Mitotic cell cycle phase transition" (Panel C) and "Anaphase-promoting complex (which of course refers to a protein assembly rather than a biological process)", "Cell division", "Mitotic cell cycle transition", "Mitotic cell cycle" and "Regulation of mitotic cell cycle" (Panel D) as separate biological processes. It would be great if you can merge some of these terms into one or two to simplify the description.